# Structured Matrix Recovery via the Generalized Dantzig Selector

**Sheng Chen**       **Arindam Banerjee**
Dept. of Computer Science & Engineering
University of Minnesota, Twin Cities
{shengc,banerjee}@cs.umn.edu

## Abstract

In recent years, structured matrix recovery problems have gained considerable attention for its real world applications, such as recommender systems and computer vision. Much of the existing work has focused on matrices with low-rank structure, and limited progress has been made on matrices with other types of structure. In this paper we present non-asymptotic analysis for estimation of generally structured matrices via the generalized Dantzig selector based on sub-Gaussian measurements. We show that the estimation error can always be succinctly expressed in terms of a few geometric measures such as Gaussian widths of suitable sets associated with the structure of the underlying true matrix. Further, we derive general bounds on these geometric measures for structures characterized by unitarily invariant norms, a large family covering most matrix norms of practical interest. Examples are provided to illustrate the utility of our theoretical development.

## 1   Introduction

Structured matrix recovery has found a wide spectrum of applications in real world, e.g., recommender systems [22], face recognition [9], etc. The recovery of an unknown structured matrix $\Theta^* \in \mathbb{R}^{d \times p}$ essentially needs to consider two aspects: the measurement model, i.e., what kind of information about the unknown matrix is revealed from each measurement, and the structure of the underlying matrix, e.g., sparse, low-rank, etc. In the context of structured matrix estimation and recovery, a widely used measurement model is the linear measurement, i.e., one has access to $n$ observations of the form $y_i = \langle\langle \Theta^*, X_i \rangle\rangle + \omega_i$ for $\Theta^*$, where $\langle\langle \cdot, \cdot \rangle\rangle$ denotes the matrix inner product, i.e., $\langle\langle A, B \rangle\rangle = \text{Tr}(A^T B)$ for any $A, B \in \mathbb{R}^{d \times p}$, and $\omega_i$'s are additive noise. In the literature, various types of measurement matrices $X_i$ has been investigated, for example, Gaussian ensemble where $X_i$ consists of i.i.d. standard Gaussian entries [11], rank-one projection model where $X_i$ is randomly generated with constraint $\text{rank}(X_i) = 1$ [7]. A special case of rank-one projection is the matrix completion model [8], in which $X_i$ has a single entry equal to 1 with all the rest set to 0, i.e., $y_i$ takes the value of one entry from $\Theta^*$ at each measurement. Other measurement models include row-and-column affine measurement [34], exponential family matrix completion [21, 20], etc.

Previous work has shown that low-complexity structure of $\Theta^*$, often captured by a small value of some norm $R(\cdot)$, can significantly benefit its recovery [11, 26]. For instance, one of the popular structures of $\Theta^*$ is low-rank, which can be approximated by a small value of trace norm (i.e., nuclear norm) $\|\cdot\|_{\text{tr}}$. Under the low-rank assumption of $\Theta^*$, recovery guarantees have been established for different measurement matrices using convex programs, e.g., trace-norm regularized least-square estimator [10, 27, 26, 21],

$$\min_{\Theta \in \mathbb{R}^{d \times p}} \ \frac{1}{2} \sum_{i=1}^{n} \left( y_i - \langle\langle X_i, \Theta \rangle\rangle \right)^2 + \beta_n \|\Theta^*\|_{\text{tr}} \,, \tag{1}$$

and constrained trace-norm minimization estimators [10, 27, 11, 7, 20], such as

$$\min_{\Theta \in \mathbb{R}^{d \times p}} \|\Theta\|_{\mathrm{tr}} \quad \text{s.t.} \quad \left\| \sum_{i=1}^{n} \left( \langle\langle X_i, \Theta \rangle\rangle - y_i \right) X_i \right\|_{\mathrm{op}} \leq \lambda_n \,, \tag{2}$$

where $\beta_n, \lambda_n$ are tuning parameters, and $\|\cdot\|_{\mathrm{op}}$ denotes the operator (spectral) norm. Among the convex approaches, the exact recovery guarantee of a matrix-form basis-pursuit [14] estimator was analyzed for the noiseless setting in [27], under certain matrix-form restricted isometry property (RIP). In the presence of noise, [10] also used matrix RIP to establish the recovery error bound for both regularized and constraint estimators, i.e., (1) and (2). In [7], a variant of estimator (2) was proposed and its recovery guarantee was built on a so-called restricted uniform boundedness (RUB) condition, which is more suitable for the rank-one projection based measurement model. Despite the fact that the low-rank structure has been well studied, only a few works extend to more general structures. In [26], the regularized estimator (1) was generalized by replacing the trace norm with a decomposable norm $R(\cdot)$ for other structures. [11] extended the estimator in [27] with $\|\cdot\|_{\mathrm{tr}}$ replaced by a norm from a broader class called atomic norm, but the consistency of the estimator is only available when the noise vector is bounded.

In this work, we make two key contributions. First, we present a general framework for estimation of structured matrices via the generalized Dantzig sector (GDS) [12, 6] as follows

$$\hat{\Theta} = \underset{\Theta \in \mathbb{R}^{d \times p}}{\operatorname{argmin}} \ R(\Theta) \quad \text{s.t.} \quad R^* \left( \sum_{i=1}^{n} \left( \langle\langle X_i, \Theta \rangle\rangle - y_i \right) X_i \right) \leq \lambda_n \,, \tag{3}$$

in which $R(\cdot)$ can be any norm and its dual norm is $R^*(\cdot)$. GDS has been studied in the context of structured vectors [12], so (3) can be viewed as a natural generalization to matrices. Note that the estimator (2) is a special case of the formulation above, as operator norm is dual to trace norm. Our deterministic analysis of the estimation error $\|\hat{\Theta} - \Theta^*\|_F$ relies on a condition based on a suitable choice of $\lambda_n$ and the restricted strong convexity (RSC) condition [26, 3]. By assuming sub-Gaussian $X_i$ and $\omega_i$, we show that these conditions are satisfied with high probability, and the recovery error can be expressed in terms of certain *geometric measures* of sets associated with $\Theta^*$. Such a geometric characterization is inspired by related advances in recent years [26, 11, 3]. One key ingredient in such characterization is the *Gaussian width* [18], which measures the size of sets in $\mathbb{R}^{d \times p}$. Related advances can be found in [11, 12, 6], but they all rely on the Gaussian measurements, to which classical concentration results [18] are directly applicable. In contrast, our work allows general sub-Gaussian measurement matrices and noise, by suitably using ideas from *generic chaining* [30], a powerful geometric approach to bounding stochastic processes. Our results can also be extended to heavy tailed measurement and noise, following recent advances [28]. Recovery guarantees of the GDS were analyzed for general norms in matrix completion setting [20], but it is different from our work since its measurement model is not sub-Gaussian as we consider.

Our second contribution is motivated by the fact that though certain existing analyses end up with the geometric measures such as Gaussian widths, limited attention has been paid in bounding these measures in terms of more easily understandable quantities especially for matrix norms. Here our key novel contribution is deriving *general* bounds for those geometric measures for the class of *unitarily invariant* norms, which are invariant under any unitary transformation, i.e., for any matrix $\Theta \in \mathbb{R}^{d \times p}$, its norm value is equal to that of $U\Theta V$ if both $U \in \mathbb{R}^{d \times d}$ and $V \in \mathbb{R}^{p \times p}$ are unitary matrices. The widely-used trace norm, spectral norm and Frobenius norm all belong to this class. A well-known result is that any unitarily invariant matrix norm is equivalent to some vector norm applied on the set of singular values [23] (see Lemma 1 for details), and this equivalence allows us to build on the techniques developed in [13] for vector norms to derive the bounds of the geometric measures for unitarily invariant norms. Previously these general bounds were not available in the literature for the matrix setting, and bounds were only in terms the geometric measures, which can be hard to interpret or bound in terms of understandable quantities. We illustrate concrete versions of the general bounds using the trace norm and the recently proposed spectral $k$-support norm [24].

The rest of the paper is organized as follows: we first provide the deterministic analysis in Section 2. In Section 3, we introduce some probability tools, which are used in the later analysis. In Section 4, we present the probabilistic analysis for sub-Gaussian measurement matrices and noise, along with the general bounds of the geometric measures for unitarily invariant norms. Section 5 is dedicated to the examples for the application of general bounds, and we conclude in Section 6.

## 2   Deterministic Recovery Guarantees

To evaluate the performance of GDS (3), we focus on the Frobenius-norm error, i.e., $\|\hat{\Theta} - \Theta^*\|_F$. Throughout the paper, w.l.o.g. we assume that $d \leq p$. For convenience, we denote the collection of $X_i$'s by $\mathbf{X} = \{X_i\}_{i=1}^n$, and let $\omega = [\omega_1, \omega_2, \ldots, \omega_n]^T$ be the noise vector. In the following theorem, we provide a deterministic bound for $\|\hat{\Theta} - \Theta^*\|_F$ under some standard assumptions on $\lambda_n$ and $\mathbf{X}$.

**Theorem 1** *Define the set* $\mathcal{E}_R(\Theta^*) = \text{cone}\{ \Delta \in \mathbb{R}^{d \times p} \mid R(\Delta + \Theta^*) \leq R(\Theta^*)\}$ *. Assume that*

$$\lambda_n \geq R^* \left( \sum_{i=1}^n \omega_i X_i \right) \; , \;\; and \;\; \sum_{i=1}^n \langle\langle X_i, \Delta \rangle\rangle^2 / \|\Delta\|_F^2 \geq \alpha > 0, \; \forall \, \Delta \in \mathcal{E}_R(\Theta^*) \, . \quad (4)$$

*Then the estimation* $\|\hat{\Theta} - \Theta^*\|_F$ *error satisfies*

$$\|\hat{\Theta} - \Theta^*\|_F \leq \frac{2\Psi_R(\Theta^*)\lambda_n}{\alpha} \, , \quad (5)$$

*where* $\Psi_R(\cdot)$ *is the restricted compatibility constant defined as* $\Psi_R(\Theta^*) = \sup_{\Delta \in \mathcal{E}_R(\Theta^*)} \frac{R(\Delta)}{\|\Delta\|_F}$.

The proof is deferred to the supplement. The convex cone $\mathcal{E}_R(\Theta^*)$ plays a important role in characterizing the error bound, and its geometry is determined by $R(\cdot)$ and $\Theta^*$. The recovery bound assumes no knowledge of the norm $R(\cdot)$ and true matrix $\Theta^*$, thus allowing general structures. The second condition in 4 is often referred to as *restricted strong convexity* [26]. In this work, we are particularly interested in $R(\cdot)$ from the class of *unitarily invariant* matrix norm, which essentially satisfies the following property, $R(\Theta) = R(U\Theta V)$ for any $\Theta \in \mathbb{R}^{d \times p}$ and unitary matrices $U \in \mathbb{R}^{d \times d}$, $V \in \mathbb{R}^{p \times p}$. A useful result for such norms is given in Lemma 1 (see [23, 4] for details).

**Lemma 1** *Suppose that the singular values of a matrix* $\Theta \in \mathbb{R}^{d \times p}$ *are given by* $\sigma = [\sigma_1, \sigma_2, \ldots, \sigma_d]^T$. *A unitarily invariant norm* $R : \mathbb{R}^{d \times p} \mapsto \mathbb{R}$ *can be characterized by some symmetric gauge function*[1] $f : \mathbb{R}^d \mapsto \mathbb{R}$ *as* $R(\Theta) = f(\sigma)$, *and its dual norm is given by* $R^*(\Theta) = f^*(\sigma)$.

As the sparsity of $\sigma$ equals the rank of $\Theta$, the class of unitarily invariant matrix norms is useful in structured low-rank matrix recovery and includes many widely used norms, e.g., trace norm with $f(\cdot) = \|\cdot\|_1$, Frobenius norm with $f(\cdot) = \|\cdot\|_2$, Schatten $p$-norm with $f(\cdot) = \|\cdot\|_p$, Ky Fan $k$-norm when $f(\cdot)$ is the $\ell_1$ norm of the largest $k$ elements in magnitude, etc.

Before proceeding with the analysis, we introduce some notations. For the rest of paper, we denote by $\sigma(\Theta) \in \mathbb{R}^d$ the vector of singular values (sorted in descending order) of matrix $\Theta \in \mathbb{R}^{d \times p}$, and may use the shorthand $\sigma^*$ for $\sigma(\Theta^*)$. For any $\theta \in \mathbb{R}^d$, we define the corresponding $|\theta|^{\downarrow}$ by arranging the absolute values of elements of $\theta$ in descending order. Given any matrix $\Theta \in \mathbb{R}^{d \times p}$ and subspace $\mathcal{M} \subseteq \mathbb{R}^{d \times p}$, we denote by $\Theta_{\mathcal{M}}$ the orthogonal projection of $\Theta$ onto $\mathcal{M}$. Besides we let $\text{colsp}(\Theta)$ ($\text{rowsp}(\Theta)$) be the subspace spanned by columns (rows) of $\Theta$. The notation $\mathbb{S}^{dp-1}$ represents the unit sphere of $\mathbb{R}^{d \times p}$, i.e., the set $\{\Theta | \|\Theta\|_F = 1\}$. The unit ball of norm $R(\cdot)$ is denoted by $\Omega_R = \{\Theta \mid R(\Theta) \leq 1\}$. Throughout the paper, the symbols $c, C, c_0, C_0$, etc., are reserved for universal constants, which may be different at each occurrence.

In the rest of our analysis, we will frequently use the so-called ordered weighted $\ell_1$ (OWL) norm for $\mathbb{R}^d$ [17], which is defined as $\|\theta\|_w \triangleq \langle |\theta|^{\downarrow}, |w|^{\downarrow}\rangle$, where $w \in \mathbb{R}^d$ is a predefined weight vector. Noting that the OWL norm is a symmetric gauge, we define the *spectral OWL norm* for $\Theta$ as: $\|\Theta\|_w \triangleq \|\sigma(\Theta)\|_w$, i.e., applying the OWL norm on $\sigma(\Theta)$.

## 3   Background and Preliminaries

The tools for our probabilistic analysis include the notion of Gaussian width [18], sub-Gaussian random matrices, and generic chaining [30]. Here we briefly introduce the basic ideas and results for each of them as needed for our analysis.

### 3.1 Gaussian width and sub-Gaussian random matrices

The Gaussian width can be defined for any subset $\mathcal{A} \subseteq \mathbb{R}^{d \times p}$ as follows [18, 19],

$$w(\mathcal{A}) \triangleq \mathbb{E}_G \Big[ \sup_{Z \in \mathcal{A}} \ \langle\langle G, Z \rangle\rangle \Big] , \tag{6}$$

where $G$ is a random matrix with i.i.d. standard Gaussian entries, i.e., $G_{ij} \sim N(0, 1)$. The Gaussian width essentially measures the size of the set $\mathcal{A}$, and some of its properties can be found in [11, 1].

A random matrix $X$ is sub-Gaussian with $\|\|X\|\|_{\psi_2} \leq \kappa$ if $\|\|\langle\langle X, Z \rangle\rangle\|\|_{\psi_2} \leq \kappa$ for any $Z \in \mathbb{S}^{dp-1}$, where the $\psi_2$ norm for sub-Gaussian random variable $x$ is defined as $\|\|x\|\|_{\psi_2} = \sup_{q \geq 1} q^{-\frac{1}{2}} (\mathbb{E}|x|^q)^{\frac{1}{q}}$ (see [31] for more details of $\psi_2$ norm). One nice property of sub-Gaussian random variable is the thin tail, i.e., $\mathbb{P}(|x| > \epsilon) \leq e \cdot \exp(-c\epsilon^2 / \|x\|_{\psi_2}^2)$, in which $c$ is a constant.

### 3.2 Generic chaining

Generic chaining is a powerful tool for bounding the supreme of stochastic processes [30]. Suppose $\{Z_t\}_{t \in \mathcal{T}}$ is a centered stochastic process, where each $Z_t$ is a centered random variable. We assume the index set $\mathcal{T}$ is endowed with some metric $s(\cdot, \cdot)$. In order to use generic chaining bound, the critical condition for $\{Z_t\}_{t \in \mathcal{T}}$ to satisfy is that, for any $u, v \in \mathcal{T}$, $\mathbb{P}(|Z_u - Z_v| \geq \epsilon) \leq c_1 \cdot \exp(-c_2 \epsilon^2 / s^2(u, v))$, where $c_1$ and $c_2$ are constants. Under this condition, we have

$$\mathbb{E}[\sup_{t \in \mathcal{T}} Z_t] \leq c_0 \gamma_2 (\mathcal{T}, s) , \tag{7}$$

$$\mathbb{P}\Big( \sup_{u,v \in \mathcal{T}} |Z_u - Z_v| \geq C_1 (\gamma_2(\mathcal{T}, s) + \epsilon \cdot \operatorname{diam}(\mathcal{T}, s)) \Big) \leq C_2 \exp(-\epsilon^2) , \tag{8}$$

where $\operatorname{diam}(\mathcal{T}, s)$ is the diameter of set $\mathcal{T}$ w.r.t. the metric $s(\cdot, \cdot)$. (7) is often referred to as generic chaining bound (see Theorem 2.2.18 and 2.2.19 in [30]), and (8) is the Theorem 2.2.27 in [30]. The functional $\gamma_2(\mathcal{T}, s)$ essentially measures the geometric size of the set $\mathcal{T}$ under the metric $s(\cdot, \cdot)$. To avoid unnecessary complications, we omit the definition of $\gamma_2(\mathcal{T}, s)$ here (see Chapter 2 of [30] for an introduction if one is interested), but provide two of its properties below,

$$\gamma_2(\mathcal{T}, s_1) \leq \gamma_2(\mathcal{T}, s_2) \ \text{if} \ s_1(u, v) \leq s_2(u, v) \ \forall \, u, v \in \mathcal{T}, \tag{9}$$

$$\gamma_2(\mathcal{T}, \eta s) = \eta \cdot \gamma_2(\mathcal{T}, s) \ \text{for any} \ \eta > 0 . \tag{10}$$

The important aspect of $\gamma_2$-functional is the following *majorizing measure theorem* [29, 30].

**Theorem 2** *Given any Gaussian process $\{Y_t\}_{t \in \mathcal{T}}$, define $s(u, v) = \sqrt{\mathbb{E}|Y_u - Y_v|^2}$ for $u, v \in \mathcal{T}$. Then $\gamma_2(\mathcal{T}, s)$ can be upper bounded by $\gamma_2(\mathcal{T}, s) \leq C_0 \mathbb{E}[\sup_{t \in \mathcal{T}} Y_t]$.*

This theorem is essentially Theorem 2.4.1 in [30]. For our purpose, we simply focus on the Gaussian process $\{Y_\Delta = \langle\langle G, \Delta \rangle\rangle\}_{\Delta \in \mathcal{A}}$, in which $\mathcal{A} \subseteq \mathbb{R}^{d \times p}$ and $G$ is a standard Gaussian random matrix. Given Theorem 2, the metric $s(U, V) = \sqrt{\mathbb{E}|\langle\langle G, U - V \rangle\rangle|^2} = \|U - V\|_F$. Therefore we have

$$\gamma_2(\mathcal{A}, \|\cdot\|_F) \leq C_0 \mathbb{E}[\sup_{\Delta \in \mathcal{A}} \langle\langle G, \Delta \rangle\rangle] = C_0 w(\mathcal{A}) , \tag{11}$$

## 4   Error Bounds with Sub-Gaussian Measurement and Noise

Though the deterministic recovery bound (5) in Section 2 applies to any measurement $\mathbf{X}$ and noise $\omega$ as long as the assumptions in (4) are satisfied, it is of practical interest to express the bound in terms of the problem parameters, e.g., $d$, $p$ and $n$, for random $\mathbf{X}$ and $\omega$ sampled from some general and widely used family of distributions. For this work, we assume that $X_i$'s in $\mathbf{X}$ are i.i.d. copies of a zero-mean random vector $X$, which is sub-Gaussian with $\|\|X\|\|_{\psi_2} \leq \kappa$ for a constant $\kappa$, and the noise $\omega$ contains i.i.d. centered random variables with $\|\omega_i\|_{\psi_2} \leq \tau$ for a constant $\tau$. In this section, we show that each quantity in (5) can be bounded using certain geometric measures associated with the true matrix $\Theta^*$. Further, we show that for unitarily invariant norms, the geometric measures can themselves be bounded in terms of $d$, $p$, $n$, and structures associated with $\Theta^*$.

## 4.1 Bounding restricted compatibility constant

Given the definition of restricted compatibility constant in Theorem 1, it involves no randomness and purely depends on $R(\cdot)$ and the geometry of $\mathcal{E}_R(\Theta^*)$. Hence we directly work on its upper bound for unitarily invariant norms. In general, characterizing the error cone $\mathcal{E}_R(\Theta^*)$ is difficult, especially for non-decomposable $R(\cdot)$. To address the issue, we first define the seminorm below.

**Definition 1** Given two orthogonal subspaces $\mathcal{M}_1, \mathcal{M}_2 \subseteq \mathbb{R}^{d \times p}$ and two vectors $w, z \in \mathbb{R}^d$, the *subspace spectral OWL seminorm* for $\mathbb{R}^{d \times p}$ is defined as $\|\Theta\|_{w,z} \triangleq \|\Theta_{\mathcal{M}_1}\|_w + \|\Theta_{\mathcal{M}_2}\|_z$, where $\Theta_{\mathcal{M}_1}$ and $\Theta_{\mathcal{M}_2}$ are the orthogonal projections of $\Theta$ onto $\mathcal{M}_1$ and $\mathcal{M}_2$, respectively.

Next we will construct such a seminorm based on a subgradient $\theta^*$ of the symmetric gauge $f$ associated with $R(\cdot)$ at $\sigma^*$, which can be obtained by solving the so-called *polar operator* [32]

$$\theta^* \in \operatorname*{argmax}_{x: f^*(x) \leq 1} \langle x, \sigma^* \rangle \ . \tag{12}$$

Given that $\sigma^*$ is sorted, w.l.o.g. we may assume that $\theta^*$ is nonnegative and sorted because $\langle \sigma^*, \theta^* \rangle \leq \langle \sigma^*, |\theta^*|^{\downarrow} \rangle$ and $f^*(\theta^*) = f^*(|\theta^*|^{\downarrow})$. Also, we denote by $\theta^*_{\max}$ ($\theta^*_{\min}$) the largest (smallest) element of the $\theta^*$, and define $\rho = \theta^*_{\max}/\theta^*_{\min}$ (if $\theta^*_{\min} = 0$, we define $\rho = +\infty$). Throughout the paper, we will frequently use these notations. As shown in the lemma below, a constructed seminorm based on $\theta^*$ will induce a set $\mathcal{E}'$ that contains $\mathcal{E}_R(\Theta^*)$ and is considerably easier to work with.

**Lemma 2** *Assume that* $\operatorname{rank}(\Theta^*) = r$ *and its compact SVD is given by* $\Theta^* = U\Sigma V^T$, *where* $U \in \mathbb{R}^{d \times r}$, $\Sigma \in \mathbb{R}^{r \times r}$ *and* $V \in \mathbb{R}^{p \times r}$. *Let* $\theta^*$ *be any subgradient of* $f(\sigma^*)$, $w = [\theta^*_1, \theta^*_2, \ldots, \theta^*_r, 0, \ldots, 0]^T \in \mathbb{R}^d$, $z = [\theta^*_{r+1}, \theta^*_{r+2}, \ldots, \theta^*_d, 0, \ldots, 0]^T \in \mathbb{R}^d$, $\mathcal{U} = \operatorname{colsp}(U)$ *and* $\mathcal{V} = \operatorname{rowsp}(V^T)$, *and define* $\mathcal{M}_1, \mathcal{M}_2$ *as* $\mathcal{M}_1 = \{\Theta \mid \operatorname{colsp}(\Theta) \subseteq \mathcal{U}, \operatorname{rowsp}(\Theta) \subseteq \mathcal{V}\}$, $\mathcal{M}_2 = \{\Theta \mid \operatorname{colsp}(\Theta) \subseteq \mathcal{U}^{\perp}, \operatorname{rowsp}(\Theta) \subseteq \mathcal{V}^{\perp}\}$, *where* $\mathcal{U}^{\perp}, \mathcal{V}^{\perp}$ *are orthogonal complements of* $\mathcal{U}$ *and* $\mathcal{V}$ *respectively. Then the specified subspace spectral OWL seminorm* $\| \cdot \|_{w,z}$ *satisfies* $\mathcal{E}_R(\Theta^*) \subseteq \mathcal{E}' \triangleq \operatorname{cone}\{\Delta \mid \|\Delta + \Theta^*\|_{w,z} \leq \|\Theta^*\|_{w,z}\}$

The proof is given in the supplementary. Base on the superset $\mathcal{E}'$, we are able to bound the restricted compatibility constant for unitarily invariant norms by the following theorem.

**Theorem 3** *Assume there exist* $\eta_1$ *and* $\eta_2$ *such that the symmetric gauge* $f$ *for* $R(\cdot)$ *satisfies* $f(\delta) \leq \max\{\eta_1\|\delta\|_1, \ \eta_2\|\delta\|_2\}$ *for any* $\delta \in \mathbb{R}^d$. *Then given a rank-$r$* $\Theta^*$, *the restricted compatibility constant* $\Psi_R(\Theta^*)$ *is upper bounded by*

$$\Psi_R(\Theta^*) \leq 2\Phi_f(r) + \max\{\eta_2, \eta_1(1 + \rho)\sqrt{r}\} \ , \tag{13}$$

*where* $\rho = \theta^*_{\max}/\theta^*_{\min}$, *and* $\Phi_f(r) = \sup_{\|\delta\|_0 \leq r} f(\delta)/\|\delta\|_2$ *is called sparse compatibility constant.*

**Remark:** The assumption for Theorem 3 might seem cumbersome at the first glance, but the different combinations of $\eta_1$ and $\eta_2$ give us more flexibility. In fact, it trivially covers two cases, $\eta_2 = 0$ along with $f(\delta) \leq \eta_1\|\delta\|_1$ for any $\delta$, and the other way around, $\eta_1 = 0$ along with $f(\delta) \leq \eta_2\|\delta\|_2$.

## 4.2 Bounding restricted convexity $\alpha$

The second condition in (4) is equivalent to $\sum_{i=1}^n \langle\langle X_i, \Delta \rangle\rangle^2 \geq \alpha > 0$, $\forall \Delta \in \mathcal{E}_R(\Theta^*) \cap \mathbb{S}^{dp-1}$. In the following theorem, we express the restricted convexity $\alpha$ in terms of Gaussian width.

**Theorem 4** *Assume that* $X_i$'s *are i.i.d. copies of a centered isotropic sub-Gaussian random matrix* $X$ *with* $\|\|X\|\|_{\psi_2} \leq \kappa$, *and let* $\mathcal{A}_R(\Theta^*) = \mathcal{E}_R(\Theta^*) \cap \mathbb{S}^{dp-1}$. *With probability at least* $1 - \exp(-\zeta w^2(\mathcal{A}_R(\Theta^*)))$, *the following inequality holds with absolute constant* $\zeta$ *and* $\xi$,

$$\inf_{\Delta \in \mathcal{A}} \frac{1}{n} \sum_{i=1}^n \langle\langle X_i, \Delta \rangle\rangle^2 \ \geq \ 1 - \xi\kappa^2 \cdot \frac{w(\mathcal{A}_R(\Theta^*))}{\sqrt{n}} \ . \tag{14}$$

The proof is essentially an application of generic chaining [30] and the following theorem from [25]. Related line of works can be found in [15, 16, 5].

**Theorem 5 (Theorem D in [25])** *There exist absolute constants $c_1$, $c_2$, $c_3$ for which the following holds. Let $(\Omega, \mu)$ be a probability space, $\mathcal{H}$ be a subset of the unit sphere of $L_2(\mu)$, i.e., $\mathcal{H} \subseteq S_{L_2} = \{h : \|\|h\|\|_{L_2} = 1\}$, and assume $\sup_{h \in \mathcal{H}} \|\|h\|\|_{\psi_2} \leq \kappa$. Then, for any $\beta > 0$ and $n \geq 1$ satisfying $c_1 \kappa \gamma_2(\mathcal{H}, \|\|\cdot\|\|_{\psi_2}) \leq \beta \sqrt{n}$, with probability at least $1 - \exp(-c_2 \beta^2 n / \kappa^4)$, we have*

$$\sup_{h \in \mathcal{H}} \left| \frac{1}{n} \sum_{i=1}^{n} h^2(X_i) - \mathbb{E}\left[h^2\right] \right| \leq \beta \ . \tag{15}$$

***Proof of Theorem 4:*** For simplicity, we use $\mathcal{A}$ as shorthand for $\mathcal{A}_R(\Theta^*)$. Let $(\Omega, \mu)$ be the probability space that $X$ is defined on, and construct

$$\mathcal{H} = \{h(\cdot) = \langle\langle \cdot, \Delta \rangle\rangle \mid \Delta \in \mathcal{A}\} \ .$$

$\|\|X\|\|_{\psi_2} \leq \kappa$ immediately implies that $\sup_{h \in \mathcal{H}} \|\|h\|\|_{\psi_2} \leq \kappa$. As $X$ is isotropic, i.e., $\mathbb{E}[\langle\langle X, \Delta \rangle\rangle^2] = 1$ for any $\Delta \in \mathcal{A} \subseteq \mathbb{S}^{dp-1}$, thus $\mathcal{H} \subseteq S_{L_2}$ and $\mathbb{E}[h^2] = 1$ for any $h \in \mathcal{H}$. Given $h_1 = \langle\langle \cdot, \Delta_1 \rangle\rangle, h_2 = \langle\langle \cdot, \Delta_2 \rangle\rangle \in \mathcal{H}$, where $\Delta_1, \Delta_2 \in \mathcal{A}$, the metric induced by $\psi_2$ norm satisfies $\|\|h_1 - h_2\|\|_{\psi_2} = \|\|\langle\langle X, \Delta_1 - \Delta_2 \rangle\rangle\|\|_{\psi_2} \leq \kappa \|\Delta_1 - \Delta_2\|_F$. Using the properties of $\gamma_2$-functional and the majorizing measure theorem in Section 3, we have

$$\gamma_2(\mathcal{H}, \|\|\cdot\|\|_{\psi_2}) \ \leq \ \kappa \gamma_2(\mathcal{A}, \|\cdot\|_F) \ \leq \ \kappa c_4 w(\mathcal{A}) \ ,$$

where $c_4$ is an absolute constant. Hence, by choosing $\beta = c_1 c_4 \kappa^2 w(\mathcal{A}) / \sqrt{n}$, we can guarantee that condition $c_1 \kappa \gamma_2(\mathcal{H}, \|\|\cdot\|\|_{\psi_2}) \leq \beta \sqrt{n}$ holds for $\mathcal{H}$. Applying Theorem 5 to this $\mathcal{H}$, with probability at least $1 - \exp(-c_2 c_1^2 c_4^2 w^2(\mathcal{A}))$, we have $\sup_{h \in \mathcal{H}} \left| \frac{1}{n} \sum_{i=1}^{n} h^2(X_i) - 1 \right| \leq \beta$, which implies

$$\inf_{\Delta \in \mathcal{A}} \frac{1}{n} \sum_{i=1}^{n} \langle\langle X_i, \Delta \rangle\rangle^2 \geq 1 - \beta \ .$$

Letting $\zeta = c_2 c_1^2 c_4^2$, $\xi = c_1 c_4$, we complete the proof. ∎

The bound (14) involves the Gaussian width of set $\mathcal{A}_R(\Theta^*)$, i.e., the error cone intersecting with unit sphere. For unitarily invariant $R$, the theorem below provides a general way to bound $w(\mathcal{A}_R(\Theta^*))$.

**Theorem 6** *Under the setting of Lemma 2, let $\rho = \theta^*_{\max} / \theta^*_{\min}$ and $\mathrm{rank}(\Theta^*) = r$. The Gaussian width $w(\mathcal{A}_R(\Theta^*))$ satisfies*

$$w(\mathcal{A}_R(\Theta^*)) \leq \min\left\{ \sqrt{dp}, \sqrt{(2\rho^2 + 1)(d + p - r)r} \right\} \ . \tag{16}$$

The proof of Theorem 6 is included in the supplementary material, which relies on a few specific properties of Gaussian random matrix [1, 11].

### 4.3 Bounding regularization parameter $\lambda_n$

In view of Theorem 1, we should choose the $\lambda_n$ large enough to satisfy the condition in (4). Hence we an upper bound for random quantity $R^*\left(\sum_{i=1}^{n} \omega_i X_i\right)$, which holds with high probability.

**Theorem 7** *Assume that $\mathbf{X} = \{X_i\}_{i=1}^{n}$ are i.i.d. copies of a centered isotropic sub-Gaussian random matrix $X$ with $\|\|X\|\|_{\psi_2} \leq \kappa$, and the noise $\omega$ consists of i.i.d. centered entries with $\|\|\omega_i\|\|_{\psi_2} \leq \tau$. Let $\Omega_R$ be the unit ball of $R(\cdot)$ and $\eta = \sup_{\Delta \in \Omega_R} \|\Delta\|_F$. With probability at least $1 - \exp(-c_1 n) - c_2 \exp\left(-w^2(\Omega_R)/c_3^2 \eta^2\right)$, the following inequality holds*

$$R^*\left(\sum_{i=1}^{n} \omega_i X_i\right) \leq c_0 \kappa \tau \cdot \sqrt{n} w(\Omega_R) \ . \tag{17}$$

*Proof:* For each entry in $\omega$, we have $\sqrt{\mathbb{E}[\omega_i^2]} \leq \sqrt{2}\|\omega_i\|_{\psi_2} = \sqrt{2}\tau$, and $\left\|\omega_i^2 - \mathbb{E}[\omega_i^2]\right\|_{\psi_1} \leq 2\left\|\omega_i^2\right\|_{\psi_1} \leq 4\|\omega_i\|_{\psi_2}^2 \leq 4\tau^2$, where we use the definition of $\psi_2$ norm and its relation to $\psi_1$ norm [31]. By Bernstein's inequality, we get

$$\mathbb{P}(\|\omega\|_2^2 - 2\tau^2 \geq \epsilon) \leq \mathbb{P}\left(\|\omega\|_2^2 - \mathbb{E}[\|\omega\|_2^2] \geq \epsilon\right) \leq \exp\left(-c_1 \min\left(\epsilon^2/16\tau^4 n,\ \epsilon/4\tau^2\right)\right) .$$

Taking $\epsilon = 4\tau^2 n$, we have $\mathbb{P}\left(\|\omega\|_2 \geq \tau\sqrt{6n}\right) \leq \exp\left(-c_1 n\right)$. Denote $Y_u = \sum_{i=1}^n u_i X_i$ for $u \in \mathbb{R}^n$. For any $u \in \mathbb{S}^{n-1}$, we get $\|Y_u\|_{\psi_2} \leq c\kappa$ due to

$$\|\langle\langle Y_u, \Delta\rangle\rangle\|_{\psi_2} = \left\|\sum_{i=1}^n u_i \langle\langle X_i, \Delta\rangle\rangle\right\|_{\psi_2} \leq c\sqrt{\sum_{i=1}^n u_i^2 \|\langle\langle X_i, \Delta\rangle\rangle\|_{\psi_2}^2} \leq c\kappa \text{ for any } \Delta \in \mathbb{S}^{dp-1}.$$

For the rest of the proof, we may drop the subscript of $Y_u$ for convenience. We construct the stochastic process $\{Z_\Delta = \langle\langle Y, \Delta\rangle\rangle\}_{\Delta \in \Omega_R}$, and note that any $Z_U$ and $Z_V$ from this process satisfy

$$\mathbb{P}\left(|Z_U - Z_V| \geq \epsilon\right) = \mathbb{P}\left(|\langle\langle Y, U - V\rangle\rangle| \geq \epsilon\right) \leq e \cdot \exp\left(-C\epsilon^2/\kappa^2 \|U - V\|_F^2\right) ,$$

for some universal constant $C$ due to the sub-Gaussianity of $Y$. As $\Omega_R$ is symmetric, it follows that

$$\sup_{U,V \in \Omega_R} |Z_U - Z_V| = 2 \sup_{\Delta \in \Omega_R} Z_\Delta , \quad \sup_{U,V \in \Omega_R} \|U - V\|_F = 2 \sup_{\Delta \in \Omega_R} \|\Delta\|_F = 2\eta .$$

Let $s(\cdot, \cdot)$ be the metric induced by norm $\kappa\|\cdot\|_F$ and $\mathcal{T} = \Omega_R$. Using deviation bound (8), we have

$$\mathbb{P}\left(2 \sup_{\Delta \in \Omega_R} Z_\Delta \geq c_4 \kappa \left(\gamma_2(\Omega_R, \|\cdot\|_F) + \epsilon \cdot 2\eta\right)\right) \leq c_2 \exp\left(-\epsilon^2\right) ,$$

where $c_2$ and $c_4$ are absolute constant. By (11), there exist constants $c_3$ and $c_5$ such that

$$\mathbb{P}\left(2R^*(Y) \geq c_5\kappa\left(w(\Omega_R) + \epsilon\right)\right) = \mathbb{P}\left(2 \sup_{\Delta \in \Omega_R} Z_\Delta \geq c_5\kappa\left(w(\Omega_R) + \epsilon\right)\right) \leq c_2 \exp\left(-\epsilon^2/c_3^2\eta^2\right) .$$

Letting $\epsilon = w(\Omega_R)$, we have $\mathbb{P}\left(R^*(Y_u) \geq c_5\kappa w(\Omega_R)\right) \leq c_2 \exp\left(-\left(w(\Omega_R)/c_3\eta\right)^2\right)$ for any $u \in \mathbb{S}^{n-1}$. Combining this with the bound for $\|\omega\|_2$ and letting $c_0 = \sqrt{6}c_5$, by union bound, we have

$$\mathbb{P}\left(R^*\left(\sum_{i=1}^n \omega_i X_i\right) \geq c_0\kappa\tau\sqrt{n}w(\Omega_R)\right) \leq \mathbb{P}\left(\frac{R^*(Y_\omega)}{\|\omega\|_2} \geq c_5\kappa w(\Omega_R)\right) + \mathbb{P}\left(\|\omega\|_2 \geq \tau\sqrt{6n}\right)$$

$$\leq \sup_{u \in \mathbb{S}^{n-1}} \mathbb{P}\left(R^*(Y_u) \geq c_5\kappa w(\Omega_R)\right) + \mathbb{P}\left(\|\omega\|_2 \geq \tau\sqrt{6n}\right) \leq c_2 \exp\left(-w^2(\Omega_R)/c_3^2\eta^2\right) + \exp\left(-c_1 n\right) ,$$

which completes the proof. ∎

The theorem above shows that the lower bound of $\lambda_n$ depends on the Gaussian width of the unit ball of $R(\cdot)$. Next we give its general bound for the unitarily invariant matrix norm.

**Theorem 8** *Suppose that the symmetric gauge $f$ associated with $R(\cdot)$ satisfies $f(\cdot) \geq \nu\|\cdot\|_1$. Then the Gaussian width $w(\Omega_R)$ is upper bounded by*

$$w(\Omega_R) \leq \frac{\sqrt{d} + \sqrt{p}}{\nu} . \tag{18}$$

## 5 Examples

Combining results in Section 4, we have that if the number of measurements $n > O(w^2(\mathcal{A}_R(\Theta^*)))$, then the recovery error, with high probability, satisfies $\|\hat{\Theta} - \Theta^*\|_F \leq O\left(\Psi_R(\Theta^*)w(\Omega_R)/\sqrt{n}\right)$. Here we give two examples based on the trace norm [10] and the recently proposed spectral $k$-support norm [24] to illustrate how to bound the geometric measures and obtain the error bound.

## 5.1 Trace norm

Trace norm has been widely used in low-rank matrix recovery. The trace norm of $\Theta^*$ is basically the $\ell_1$ norm of $\sigma^*$, i.e., $f = \|\cdot\|_1$. Now we turn to the three geometric measures. Assuming that $\mathrm{rank}(\Theta^*) = r \ll d$, one subgradient of $\|\sigma^*\|_1$ is $\theta^* = [1, 1, \ldots, 1]^T$.

**Restricted compatibility constant** $\Psi_{\mathrm{tr}}(\Theta^*)$: It is obvious that assumption in Theorem 3 will hold for $f$ by choosing $\eta_1 = 1$ and $\eta_2 = 0$, and we have $\rho = 1$. The sparse compatibility constant $\Phi_{\ell_1}(r)$ is $\sqrt{r}$ because $\|\delta\|_1 \le \sqrt{r}\|\delta\|_2$ for any $r$-sparse $\delta$. Using Theorem 3, we have $\Psi_{\mathrm{tr}}(\Theta^*) \le 4\sqrt{r}$.

**Gaussian width** $w(\mathcal{A}_{\mathrm{tr}}(\Theta^*))$: As $\rho = 1$, Theorem 6 implies that $w(\mathcal{A}_{\mathrm{tr}}(\Theta^*)) \le \sqrt{3r(d + p - r)}$.

**Gaussian width** $w(\Omega_{\mathrm{tr}})$: Using Theorem 8 with $\nu = 1$, it is easy to see that $w(\Omega_{\mathrm{tr}}) \le \sqrt{d} + \sqrt{p}$.

Putting all the results together, we have $\|\hat{\Theta} - \Theta^*\|_F \le O(\sqrt{rd/n} + \sqrt{rp/n})$ holds with high probability when $n > O(r(d + p - r))$, which matches the bound in [8].

## 5.2 Spectral $k$-support norm

The $k$-support norm proposed in [2] is defined as

$$\|\theta\|_k^{sp} \triangleq \inf\left\{\sum_i \|u_i\|_2 \ \Big|\ \|u_i\|_0 \le k,\ \sum_i u_i = \theta\right\}, \tag{19}$$

and its dual norm is simply given by $\|\theta\|_k^{sp*} = \|\theta|_{1:k}^{\downarrow}\|_2$. It is shown that $k$-support norm has similar behavior as elastic-net regularizer [33]. Spectral $k$-support norm (denoted by $\|\cdot\|_{\mathrm{sk}}$) of $\Theta^*$ is defined by applying the $k$-support norm on $\sigma^*$, i.e., $f = \|\cdot\|_k^{sp}$, which has demonstrated better performance than trace norm in matrix completion task [24]. For simplicity, We assume that $\mathrm{rank}(\Theta^*) = r = k$ and $\|\sigma^*\|_2 = 1$. One subgradient of $\|\sigma^*\|_k^{sp}$ can be $\theta^* = [\sigma_1^*, \sigma_2^*, \ldots, \sigma_r^*, \sigma_r^*, \ldots, \sigma_r^*]^T$.

**Restricted compatibility constant** $\Psi_{\mathrm{sk}}(\Theta^*)$: The following relation has been shown for $k$-support norm in [2],

$$\max\{\|\cdot\|_2, \|\cdot\|_1/\sqrt{k}\} \le \|\cdot\|_k^{sp} \le \sqrt{2}\max\{\|\cdot\|_2, \|\cdot\|_1/\sqrt{k}\}. \tag{20}$$

Hence the assumption in Theorem 3 will hold for $\eta_1 = \sqrt{\frac{2}{k}}$ and $\eta_2 = \sqrt{2}$, and we have $\rho = \sigma_1^*/\sigma_r^*$. The sparse compatibility constant $\Phi_k^{sp}(r) = \Phi_k^{sp}(k) = 1$ because $\|\delta\|_k^{sp} = \|\delta\|_2$ for any $k$-sparse $\delta$. Using Theorem 3, we have $\Psi_{\mathrm{sk}}(\Theta^*) \le 2\sqrt{2} + \sqrt{2}\left(1 + \sigma_1^*/\sigma_r^*\right) = \sqrt{2}\left(3 + \sigma_1^*/\sigma_r^*\right)$.

**Gaussian width** $w(\mathcal{A}_{\mathrm{sk}}(\Theta^*))$: Theorem 6 implies $w(\mathcal{A}_{\mathrm{sk}}(\Theta^*)) \le \sqrt{r(d + p - r)\left[2\sigma_1^{*2}/\sigma_r^{*2} + 1\right]}$.

**Gaussian width** $w(\Omega_{\mathrm{sk}})$: The relation above for $k$-support norm shown in [2] also implies that $\nu = 1/\sqrt{k} = 1/\sqrt{r}$. By Theorem 8, we get $w(\Omega_{\mathrm{sk}}) \le \sqrt{r}(\sqrt{d} + \sqrt{p})$.

Given the upper bounds for geometric measures, with high probability, we have $\|\hat{\Theta} - \Theta^*\|_F \le O(\sqrt{rd/n} + \sqrt{rp/n})$ when $n > O(r(d + p - r))$. The spectral $k$-support norm was first introduced in [24], in which no statistical results are provided. Although [20] investigated the statistical aspects of spectral $k$-support norm in matrix completion setting, the analysis was quite different from our setting. Hence this error bound is new in the literature.

## 6 Conclusions

In this work, we present the recovery analysis for matrices with general structures, under the setting of sub-Gaussian measurement and noise. Base on generic chaining and Gaussian width, the recovery guarantees can be succinctly summarized in terms of some geometric measures. For the class of unitarily invariant norms, we also provide novel general bounds of these measures, which can significantly facilitate the analysis in future.

## Acknowledgements

The research was supported by NSF grants IIS-1563950, IIS-1447566, IIS-1447574, IIS-1422557, CCF-1451986, CNS- 1314560, IIS-0953274, IIS-1029711, NASA grant NNX12AQ39A, and gifts from Adobe, IBM, and Yahoo.

## Footnotes

[1]Symmetric gauge function is a norm that is invariant under sign-changes and permutations of the elements.

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
