[Supplementary Material]

# Supplementary Material to Structured Matrix Recovery via the Generalized Dantzig Selector

**Sheng Chen**     **Arindam Banerjee**
Dept. of Computer Science & Engineering
University of Minnesota, Twin Cities
{shengc,banerjee}@cs.umn.edu

## 1 Proof of Theorem 1

**Statement of Theorem:** *Define the set $\mathcal{E}_R(\Theta^*) = \text{cone}\{\Delta \mid R(\Delta + \Theta^*) \leq R(\Theta^*)\}$. Assume the following conditions hold for $\lambda_n$ and $\mathbf{X}$,*

$$\lambda_n \geq R^* \left( \sum_{i=1}^{n} \omega_i X_i \right) , \tag{S.1}$$

$$\sum_{i=1}^{n} \langle\langle X_i, \Delta \rangle\rangle^2 / \|\Delta\|_F^2 \geq \alpha > 0, \ \forall \ \Delta \in \mathcal{E}_R(\Theta^*) . \tag{S.2}$$

*The estimation $\|\hat{\Theta} - \Theta^*\|_F$ error satisfies*

$$\|\hat{\Theta} - \Theta^*\|_F \leq \frac{2\Psi_R(\Theta^*) \cdot \lambda_n}{\alpha} , \tag{S.3}$$

*where $\Psi_R(\cdot)$ is the restricted compatibility constant defined as*

$$\Psi_R(\Theta^*) = \sup_{\Delta \in \mathcal{E}_R(\Theta^*)} \frac{R(\Delta)}{\|\Delta\|_F} \tag{S.4}$$

*Proof:*    Since $\lambda_n$ satisfies the condition (S.1) and $\omega_i = y_i - \langle\langle X_i, \Theta^* \rangle\rangle$, we have

$$R^* \left( \sum_{i=1}^{n} \left( \langle\langle X_i, \Theta^* \rangle\rangle - y_i \right) X_i \right) \leq \lambda_n ,$$

which indicates that the constraint set in (3) is feasible, thus

$$R^* \left( \sum_{i=1}^{n} \left( \langle\langle X_i, \hat{\Theta} \rangle\rangle - y_i \right) X_i \right) \leq \lambda_n .$$

Using triangular inequality, one has

$$R^* \left( \sum_{i=1}^{n} \langle\langle X_i, \hat{\Theta} - \Theta^* \rangle\rangle \cdot X_i \right) \leq 2\lambda_n .$$

Denote $\hat{\Theta} - \Theta^*$ by $\Delta$, and by the definition of dual norm, we get

$$\sum_{i=1}^{n} \langle\langle X_i, \Delta \rangle\rangle^2 = \langle\langle \Delta, \sum_{i=1}^{n} \langle\langle X_i, \Delta \rangle\rangle \cdot X_i \rangle\rangle \leq R(\Delta) \cdot R^* \left( \sum_{i=1}^{n} \langle\langle X_i, \hat{\Theta} - \Theta^* \rangle\rangle \cdot X_i \right) \leq 2\lambda_n R(\Delta) .$$

On the other hand, the objective function in (3) implies that $R(\hat{\Theta}) \le R(\Theta^*)$. Therefore the error vector $\Delta$ must belong to the set $\mathcal{E}_R(\Theta^*)$. Using condition (S.2), we obtain

$$\alpha\|\Delta\|_F^2 \le \sum_{i=1}^{n} \langle\langle X_i, \Delta\rangle\rangle^2 \le 2\lambda_n R(\Delta) \,,$$

$$\|\Delta\|_F \le \frac{2\lambda_n}{\alpha} \frac{R(\Delta)}{\|\Delta\|_F} \le \frac{2\Psi_R(\Theta^*) \cdot \lambda_n}{\alpha} \,,$$

which complete the proof. ∎

## 2 Proof of Lemma 2

**Statement of Lemma:** *Assume that* $\mathrm{rank}(\Theta^*) = r$ *and its compact SVD is given by* $\Theta^* = U\Sigma V^T$, *where* $U \in \mathbb{R}^{d \times r}$, $\Sigma \in \mathbb{R}^{r \times r}$ *and* $V \in \mathbb{R}^{p \times r}$. *Let* $\theta^*$ *be any subgradient of* $f(\sigma^*)$, $w = [\theta_1^*, \theta_2^*, \ldots, \theta_r^*, 0, \ldots, 0]^T \in \mathbb{R}^d$, $z = [\theta_{r+1}^*, \theta_{r+2}^*, \ldots, \theta_d^*, 0, \ldots, 0]^T \in \mathbb{R}^d$, $\mathcal{U} = \mathrm{colsp}(U)$ *and* $\mathcal{V} = \mathrm{rowsp}(V^T)$, *and define* $\mathcal{M}_1$, $\mathcal{M}_2$ *as*

$$\mathcal{M}_1 = \{\Theta \mid \mathrm{colsp}(\Theta) \subseteq \mathcal{U}, \mathrm{rowsp}(\Theta) \subseteq \mathcal{V}\} \,,$$
$$\mathcal{M}_2 = \{\Theta \mid \mathrm{colsp}(\Theta) \subseteq \mathcal{U}^\perp, \mathrm{rowsp}(\Theta) \subseteq \mathcal{V}^\perp\} \,,$$

*where* $\mathcal{U}^\perp$, $\mathcal{V}^\perp$ *are orthogonal complements of* $\mathcal{U}$ *and* $\mathcal{V}$ *respectively. Then the specified subspace spectral OWL seminorm* $\|\cdot\|_{w,z}$ *satisfies*

$$\mathcal{E}_R(\Theta^*) \subseteq \mathcal{E}' \triangleq \mathrm{cone}\{\Delta \mid \|\Delta + \Theta^*\|_{w,z} \le \|\Theta^*\|_{w,z}\}$$

*Proof:* Both $\mathcal{E}_R(\Theta^*)$ and $\mathcal{E}'$ are induced by scaled (semi)norm balls (i.e., $\Omega_R$ and $\Omega_{w,z}$) centered at $-\Theta^*$, and note that

$$\Theta^*_{\mathcal{M}_1} = \Theta^* \,, \quad \Theta^*_{\mathcal{M}_2} = 0 \,.$$

Thus we obtain

$$\|\Theta^*\|_{w,z} = \|\Theta^*_{\mathcal{M}_1}\|_w = \sum_{i=1}^{r} \sigma_i^* \theta_i^* = \langle\sigma^*, \theta^*\rangle = R(\Theta^*) \,,$$

which indicates that the two balls have the same radius. Hence we only need to show that $\|\cdot\|_{w,z} \le R(\cdot)$. For any $\Delta \in \mathbb{R}^{d \times p}$, assume that the SVD of $\Delta_{\mathcal{M}_1}$ and $\Delta_{\mathcal{M}_2}$ are given by $\Delta_{\mathcal{M}_1} = U_1\Sigma_1 V_1^T$ and $\Delta_{\mathcal{M}_2} = U_2\Sigma_2 V_2^T$. The corresponding vectors of singular values are in the form of $\sigma' = [\sigma_1', \sigma_2', \ldots, \sigma_r', 0, \ldots, 0]^T, \sigma'' = [\sigma_1'', \sigma_2'', \ldots, \sigma_{d-r}'', 0, \ldots, 0]^T \in \mathbb{R}^d$, as $\mathrm{rank}(\Delta_{\mathcal{M}_1}) \le r$ and $\mathrm{rank}(\Delta_{\mathcal{M}_2}) \le d - r$. Then we have

$$\|\Delta\|_{w,z} = \|\Delta_{\mathcal{M}_1}\|_w + \|\Delta_{\mathcal{M}_2}\|_z = \langle\sigma', w\rangle + \langle\sigma'', z\rangle = \left\langle \theta^*, \begin{bmatrix} \sigma'_{1:r} \\ \sigma''_{1:d-r} \end{bmatrix} \right\rangle = \langle\langle\Theta, \Delta\rangle\rangle \,,$$

where $\Theta = U_1 \mathrm{Diag}(\theta^*_{1:r})V_1 + U_2 \mathrm{Diag}(\theta^*_{r+1:n})V_2$. From this construction, we can see that $\theta^*$ are the singular values of $\Theta$, thus $R^*(\Theta) \le 1$. It follows that

$$\langle\langle\Theta, \Delta\rangle\rangle \le \max_{R^*(Z) \le 1} \langle\langle Z, \Delta\rangle\rangle = R(\Delta) \,,$$

which completes the proof. ∎

## 3 Proof of Theorem 3

**Statement of Theorem:** *Assume there exist* $\eta_1$ *and* $\eta_2$ *such that the symmetric gauge* $f$ *associated with* $R(\cdot)$ *satisfies*

$$f(\delta) \le \max\{\eta_1\|\delta\|_1, \, \eta_2\|\delta\|_2\} \tag{S.5}$$

*for any $\delta \in \mathbb{R}^d$. Then given a rank-$r$ $\Theta^*$, the restricted compatibility constant $\Psi_R(\Theta^*)$ can be upper bounded by*

$$\Psi_R(\Theta^*) \leq 2\Phi_f(r) + \max\left\{\eta_2, \eta_1(1+\rho)\sqrt{r}\right\}, \tag{S.6}$$

*where $\Phi_f(r) = \sup_{\|\delta\|_0 \leq r} \frac{f(\delta)}{\|\delta\|_2}$ is called sparse compatibility constant.*

*Proof:* Under the setting of Lemma 2, as $\Theta^* \in \mathcal{M}_1$, we have

$$\|\Delta + \Theta^*\|_{w,z} \leq \|\Theta^*\|_{w,z} \implies \|\Delta_{\mathcal{M}_1} + \Theta^*\|_w + \|\Delta_{\mathcal{M}_2}\|_z \leq \|\Theta^*\|_w \implies$$
$$-\|\Delta_{\mathcal{M}_1}\|_w + \|\Theta^*\|_w + \|\Delta_{\mathcal{M}_2}\|_z \leq \|\Theta^*\|_w \implies \|\Delta_{\mathcal{M}_2}\|_z \leq \|\Delta_{\mathcal{M}_1}\|_w .$$

As the set $\{\Delta \mid \|\Delta_{\mathcal{M}_2}\|_z \leq \|\Delta_{\mathcal{M}_1}\|_w\}$ itself is a cone, we obtain

$$\mathcal{E}' \subseteq \{\Delta \mid \|\Delta_{\mathcal{M}_2}\|_z \leq \|\Delta_{\mathcal{M}_1}\|_w\}$$

Define $\mathcal{M}^\perp$ as the orthogonal complement of $\mathcal{M}_1 \oplus \mathcal{M}_2$. By the definition and Lemma 2, we have

$$\Psi_R(\Theta^*) = \sup_{\Delta \in \mathcal{E}_R(\Theta^*)} \frac{R(\Delta)}{\|\Delta\|_F} \leq \sup_{\Delta \in \mathcal{E}'} \frac{R(\Delta)}{\|\Delta\|_F} \leq \sup_{\|\Delta_{\mathcal{M}_2}\|_z \leq \|\Delta_{\mathcal{M}_1}\|_w} \frac{R(\Delta)}{\|\Delta\|_F}$$

$$\leq \sup_{\|\Delta_{\mathcal{M}_2}\|_z \leq \|\Delta_{\mathcal{M}_1}\|_w} \frac{R(\Delta_{\mathcal{M}^\perp}) + R(\Delta_{\mathcal{M}_1} + \Delta_{\mathcal{M}_2})}{\|\Delta\|_F}$$

$$\leq \sup_{\Delta \in \mathcal{M}^\perp} \frac{R(\Delta)}{\|\Delta\|_F} + \sup_{\frac{\|\Delta_{\mathcal{M}_2}\|_{\mathrm{tr}}}{\|\Delta_{\mathcal{M}_1}\|_{\mathrm{tr}}} \leq \rho} \frac{R(\Delta_{\mathcal{M}_1} + \Delta_{\mathcal{M}_2})}{\|\Delta\|_F}$$

It is not difficult to see that any $\Delta \in \mathcal{M}^\perp$ has rank at most $2r$, thus

$$\sup_{\Delta \in \mathcal{M}^\perp} \frac{R(\Delta)}{\|\Delta\|_F} = \sup_{\Delta \in \mathcal{M}^\perp} \frac{f(\sigma(\Delta))}{\|\sigma(\Delta)\|_2} \leq \sup_{\|\delta\|_0 \leq 2r} \frac{f(\delta)}{\|\delta\|_2} \leq 2\sup_{\|\delta\|_0 \leq r} \frac{f(\delta)}{\|\delta\|_2} = 2\Phi_f(r) .$$

Using (S.5) and $\|\Delta_{\mathcal{M}_1} + \Delta_{\mathcal{M}_2}\|_F \leq \|\Delta\|_F$, we have

$$\sup_{\frac{\|\Delta_{\mathcal{M}_2}\|_{\mathrm{tr}}}{\|\Delta_{\mathcal{M}_1}\|_{\mathrm{tr}}} \leq \rho} \frac{R(\Delta_{\mathcal{M}_1} + \Delta_{\mathcal{M}_2})}{\|\Delta\|_F} \leq \sup_{\frac{\|\Delta_{\mathcal{M}_2}\|_{\mathrm{tr}}}{\|\Delta_{\mathcal{M}_1}\|_{\mathrm{tr}}} \leq \rho} \frac{\max\left\{\eta_2\|\Delta\|_F, \eta_1\|\Delta_{\mathcal{M}_1} + \Delta_{\mathcal{M}_2}\|_{\mathrm{tr}}\right\}}{\|\Delta\|_F}$$

$$\leq \max\left\{\eta_2, \sup_{\Delta \in \mathcal{M}_1} \frac{\eta_1(1+\rho)\|\Delta\|_{\mathrm{tr}}}{\|\Delta\|_F}\right\}$$

$$\leq \max\left\{\eta_2, \eta_1(1+\rho)\sqrt{r}\right\} ,$$

where the last inequality uses the fact that any $\Delta \in \mathcal{M}_1$ is at most rank-$r$, and $\|\delta\|_1 \leq \sqrt{r}\|\delta\|_2$ for any $r$-sparse vector $\delta$. Combining all the inequalities, we complete the proof. ∎

## 4 Properties of Gaussian Random Matrix

To facilitate the computation of Gaussian width, especially the proof of Theorem 6, we will use some properties specific to the Gaussian random matrix $G \in \mathbb{R}^{d \times p}$, which are summarized as follows. The symbol "$\sim$" means "has the same distribution as".

**Property 1:** Given an $m$-dimensional subspace $\mathcal{M} \subseteq \mathbb{R}^{d \times p}$ spanned by orthonormal basis $U_1, \ldots, U_m$,

$$G_{\mathcal{M}} \sim \sum_{i=1}^{m} g_i U_i,$$

where $g_i$'s are i.i.d. standard Gaussian random variables. Moreover, $\mathbb{E}\left[\|G_{\mathcal{M}}\|_F^2\right] = m$.

*Proof:* Given the orthonormal basis $U_1, \ldots, U_m$ of subspace $\mathcal{M}$, $G_{\mathcal{M}}$ can be written as

$$G_{\mathcal{M}} = \sum_{i=1}^{m} \langle\langle G, U_i \rangle\rangle \cdot U_i$$

Since $\|U_1\|_F = \ldots = \|U_m\|_F = 1$, each $\langle\langle G, U_i \rangle\rangle$ is standard Gaussian. Moreover, as $U_1, \ldots, U_m$ are orthogonal, $\langle\langle G, U_i \rangle\rangle$ are independent of each other. ∎

**Property 2:** $G_{\mathcal{M}_1}$ and $G_{\mathcal{M}_2}$ are independent if $\mathcal{M}_1, \mathcal{M}_2 \subseteq \mathbb{R}^{d \times p}$ are orthogonal subspaces.

*Proof:* Suppose that the orthonormal bases of $\mathcal{M}_1, \mathcal{M}_2$ are given by $U_1, \ldots, U_{m_1}$ and $V_1, \ldots, V_{m_2}$ respectively. Using Property 1 above, $G_{\mathcal{M}_1}$ and $G_{\mathcal{M}_2}$ can be written as

$$G_{\mathcal{M}_1} = \sum_{i=1}^{m_1} \langle\langle G, U_i \rangle\rangle \cdot U_i ~\sim~ \sum_{i=1}^{m_1} g_i U_i \,,$$

$$G_{\mathcal{M}_2} = \sum_{i=1}^{m_2} \langle\langle G, V_i \rangle\rangle \cdot V_i ~\sim~ \sum_{i=1}^{m_2} h_i V_i \,,$$

where $g_1, \ldots, g_{m_1}$ and $h_1, \ldots, h_{m_2}$ are all standard Gaussian. As $\mathcal{M}_1, \mathcal{M}_2 \subseteq \mathbb{R}^{d \times p}$ are orthogonal, $U_1, \ldots, U_{m_1}$ and $V_1, \ldots, V_{m_2}$ are orthogonal to each other as well, which implies that $g_1, \ldots, g_{m_1}$ and $h_1, \ldots, h_{m_2}$ are all independent. Therefore $G_{\mathcal{M}_1}$ and $G_{\mathcal{M}_2}$ are independent. ∎

**Property 3:** Given a subspace

$$\mathcal{M} = \{\Theta \in \mathbb{R}^{d \times p} \mid \mathrm{colsp}(\Theta) \subseteq \mathcal{U}, ~ \mathrm{rowsp}(\Theta) \subseteq \mathcal{V}\} \,,$$

where $\mathcal{U} \subseteq \mathbb{R}^d$, $\mathcal{V} \subseteq \mathbb{R}^p$ are two subspaces of dimension $m_1$ and $m_2$ respectively, then $\|G_{\mathcal{M}}\|_{\mathrm{op}}$ satisfies

$$\|G_{\mathcal{M}}\|_{\mathrm{op}} \sim \|G'\|_{\mathrm{op}} \,,$$

where $G'$ is an $m_1 \times m_2$ matrix with i.i.d. standard Gaussian entries.

*Proof:* Suppose that the orthonormal bases for $\mathcal{U}$ and $\mathcal{V}$ are $U = [u_1, \ldots, u_{m_1}]$ and $V = [v_1, \ldots, v_{m_2}]$ respectively, and $U_\perp$ and $V_\perp$ denote the orthonormal bases for their orthogonal complement. It is easy to see that the orthonormal basis for $\mathcal{M}$ can be given by $\{u_i v_j^T \mid 1 \leq i \leq m_1, ~ 1 \leq j \leq m_2\}$. Using Property 1, we have

$$G_{\mathcal{M}} ~\sim~ \sum_{i=1}^{m_1} \sum_{j=1}^{m_2} g'_{ij} u_i v_j^T = U G' V = [U, U_\perp] \cdot \begin{bmatrix} G' & 0_{m_1 \times (p-m_2)} \\ 0_{(d-m_1) \times m_2} & 0_{(d-m_1) \times (p-m_2)} \end{bmatrix} \cdot \begin{bmatrix} V^T \\ V_\perp^T \end{bmatrix}$$

where $G'$ is a $m_1 \times m_2$ standard Gaussian random matrix. Note that both $[U, U_\perp] \in \mathbb{R}^{d \times d}$ and $[V, V_\perp] \in \mathbb{R}^{p \times p}$ are unitary matrices, because they form the orthonormal bases for $\mathbb{R}^d$ and $\mathbb{R}^p$ respectively. If we denote $\begin{bmatrix} G' & 0 \\ 0 & 0 \end{bmatrix}$ by $W$, then $\|G_{\mathcal{M}}\|_{\mathrm{op}} = \|W\|_{\mathrm{op}}$ as spectral norm is unitarily invariant. Further, if the SVD of $G'$ is $G' = U_1 \Sigma_1 V_1^T$, where $U_1 \in \mathbb{R}^{m_1 \times m_1}$, $\Sigma_1 \in \mathbb{R}^{m_1 \times m_2}$ and $V_1 \in \mathbb{R}^{m_2 \times m_2}$, then the SVD of $W$ is given by

$$W = \begin{bmatrix} U_1 & 0_{m_1 \times (d-m_1)} \\ 0_{(d-m_1) \times m_1} & U_2 \end{bmatrix} \begin{bmatrix} \Sigma_1 & 0_{m_1 \times (p-m_2)} \\ 0_{(d-m_1) \times m_2} & 0_{(d-m_1) \times (p-m_2)} \end{bmatrix} \begin{bmatrix} V_1^T & 0_{m_2 \times (p-m_2)} \\ 0_{(p-m_2) \times m_2} & V_2^T \end{bmatrix} \,,$$

where $U_2 \in \mathbb{R}^{(d-m_1) \times (d-m_1)}$ and $V_2 \in \mathbb{R}^{(p-m_2) \times (p-m_2)}$ are arbitrary unitary matrices. From the equation above, we can see that $W$ and $G'$ share the same singular values, thus $\|G_{\mathcal{M}}\|_{\mathrm{op}} = \|W\|_{\mathrm{op}} = \|G'\|_{\mathrm{op}}$. ∎

**Property 4:** The operator norm $\|G\|_{\mathrm{op}}$ satisfies

$$\mathbb{P}\left(\|G\|_{\mathrm{op}} \geq \sqrt{d} + \sqrt{p} + \epsilon\right) \leq \exp\left(-\frac{\epsilon^2}{2}\right) \,, \tag{S.7}$$

$$\mathbb{E}\left[\|G\|_{\mathrm{op}}\right] \leq \sqrt{d} + \sqrt{p} \,, \tag{S.8}$$

$$\mathbb{E}\left[\|G\|_{\mathrm{op}}^2\right] \leq \left(\sqrt{d} + \sqrt{p}\right)^2 + 2 \,. \tag{S.9}$$

(S.7) and (S.8) are the classical results on the extreme singular value of Gaussian random matrix [4, 5] (see Theorem 5.32 and Corollary 5.35 in [5]). (S.9) is used in [2] (see (82) - (87) in [2]).

**Property 5:** For a subset of unit sphere $\mathcal{A} \subseteq \mathbb{S}^{dp-1}$, A useful inequality [2, 1] is given by the Gaussian width satisfies

$$w^2(\mathcal{A}) \le \mathbb{E}_G[\inf_{Z \in \mathcal{N}} \|G - Z\|_F^2] , \tag{S.10}$$

in which $\mathcal{N} = \{Z \mid \langle\langle Z, \Delta \rangle\rangle \le 0 \text{ for all } \Delta \in \mathcal{A}\}$ is the polar cone of $\mathrm{cone}(\mathcal{A})$.

This property is essentially Proposition 10.2 in [1], and the right-hand side is often called *statistical dimension*.

## 5  Proof of Theorem 6

**Statement of Theorem:** *Under the setting of Lemma 2, let $\rho = \theta^*_{\max}/\theta^*_{\min}$ and $\mathrm{rank}(\Theta^*) = r$. The Gaussian width $w(\mathcal{A}_R(\Theta^*))$ satisfies*

$$w(\mathcal{A}_R(\Theta^*)) \le \min \left\{ \sqrt{dp}, \sqrt{(2\rho^2 + 1)(d + p - r)\, r} \right\} . \tag{S.11}$$

*Proof:*  For simplicity, we use $\mathcal{A}$ as shorthand for $\mathcal{A}_R(\Theta^*)$. Let $\theta^*$ be any subgradient of $f(\cdot)$ at $\sigma^*$, i.e., $\theta^* \in \partial f(\sigma^*)$, and $\Gamma = U\,\mathrm{Diag}(\theta^*_{1:r})V$. We define

$$\mathcal{D} = \{W \mid W \in \mathcal{M}_2,\ \sigma(W) \preceq z\} , \quad \mathcal{K} = \{\Gamma + W \mid W \in \mathcal{D}\} ,$$

where the symbol "$\preceq$" means "elementwise less than or equal". It is not difficult to see that $\mathcal{K}$ is a subset of $\partial R(\Theta^*)$, as any $Z \in \mathcal{K}$ satisfies $R^*(Z) = f^*(\sigma(Z)) \le f^*(\theta^*) = 1$ and $\langle\langle Z, \Theta^* \rangle\rangle = \langle \sigma(Z), \sigma^* \rangle = \langle \theta^*_{1:r}, \sigma^*_{1:r} \rangle = f(\sigma^*) = R(\Theta^*)$. Hence we have

$$\mathrm{cone}(\mathcal{K}) \subset \mathrm{cone}\{\partial R(\Theta^*)\} = \mathcal{N} ,$$

where $\mathcal{N}$ is the polar cone of $\mathcal{E}_R(\Theta^*)$, and the equality follows from the Theorem 23.7 of [3]. We define the subspace $\mathcal{M}^\perp$ as the orthogonal complement of $\mathcal{M}_1 \oplus \mathcal{M}_2$. For the sake of convenience, we denote by $G_1$ ($G_2$, $G_\perp$) the orthogonal projection of $G$ onto $\mathcal{M}_1$ ($\mathcal{M}_2$, $\mathcal{M}_\perp$), and denote $\mathrm{cone}(\mathcal{K})$ by $\mathcal{C}$. Using (S.10), we obtain

$$w(\mathcal{A})^2 \le \mathbb{E}\left[\inf_{Z \in \mathcal{N}} \|G - Z\|_F^2\right] \le \mathbb{E}\left[\inf_{Z \in \mathcal{C}} \|G_1 - Z_1\|_F^2 + \|G_2 - Z_2\|_F^2 + \|G_\perp - Z_\perp\|_F^2\right]$$
$$= \mathbb{E}\left[\inf_{t \ge 0,\ W \in t\mathcal{D}} \|G_1 - t\Gamma\|_F^2 + \|G_2 - W\|_F^2\right] + \mathbb{E}\left[\|G_\perp\|_F^2\right] . \tag{S.12}$$

To further bound the expectations, we let $t_0 = \|G_2\|_{\mathrm{op}}/\theta^*_{\min}$, which is a random quantity depending on $G$. Therefore, we have

$$\mathbb{E}\left[\inf_{t \ge 0,\ W \in t\mathcal{D}} \|G_1 - t\Gamma\|_F^2 + \|G_2 - W\|_F^2\right] \le \mathbb{E}\left[\|G_1 - t_0\Gamma\|_F^2\right] + \mathbb{E}\left[\inf_{W \in t_0\mathcal{D}} \|G_2 - W\|_F^2\right]$$
$$= \mathbb{E}\left[\|G_1\|_F^2\right] + 2\mathbb{E}\left[\langle\langle G_1, t_0\Gamma \rangle\rangle\right] + \|\theta^*_{1:r}\|_2^2 \cdot \mathbb{E}\left[t_0^2\right] + 0$$
$$= r^2 + 0 + \mathbb{E}\left[\|G_2\|_{\mathrm{op}}^2\right] \cdot \|\theta^*_{1:r}\|_2^2/\theta^{*2}_{\min}$$
$$\le r^2 + ((\sqrt{d-r} + \sqrt{p-r})^2 + 2) \cdot \|\theta^*_{1:r}\|_2^2/\theta^{*2}_{\min}$$
$$\le r^2 + 2\rho^2 r\,(d + p - 2r) , \tag{S.13}$$

where the second equality uses Property 1 and 2 in Section 4, and the second inequality follows from Property 3 and 4. Since $\mathcal{M}_\perp$ is a $r(d + p - 2r)$-dimensional subspace, by Property 1 we have $\mathbb{E}\left[\|G_\perp\|_F^2\right] = r(d + p - 2r)$. Combining it with (S.12) and (S.13), we have $w(\mathcal{A}) \le \sqrt{(2\rho^2 + 1)(d + p - r)\, r}$. On the other hand, as $\mathcal{A} \subseteq \mathbb{S}^{dp-1}$, we always have $w(\mathcal{A}) \le \mathbb{E}[\|G\|_F] \le \sqrt{\mathbb{E}[\|G\|_F^2]} = \sqrt{dp}$. We finish the proof by combining the two bounds for $w(\mathcal{A})$.  ∎

## 6  Proof of Theorem 8

**Statement of Theorem:** *Suppose that the symmetric gauge $f$ associated with $R(\cdot)$ satisfies $f(\cdot) \geq \nu \| \cdot \|_1$. Then the Gaussian width $w(\Omega_R)$ is upper bounded by*

$$w(\Omega_R) \leq \frac{\sqrt{d} + \sqrt{p}}{\nu} \tag{S.14}$$

*Proof:*  As $f(\cdot) \geq \nu \| \cdot \|_1$, we have

$$R(\cdot) \geq \nu \| \cdot \|_{\mathrm{tr}} \quad \Longrightarrow \quad \Omega_R \subseteq \Omega_{\nu \| \cdot \|_{\mathrm{tr}}} .$$

Hence it follows that

$$w\left(\Omega_R\right) \leq w\left(\Omega_{\nu \| \cdot \|_{\mathrm{tr}}}\right) = \frac{w\left(\Omega_{\| \cdot \|_{\mathrm{tr}}}\right)}{\nu} = \frac{\mathbb{E}\|G\|_{\mathrm{op}}}{\nu} \leq \frac{\sqrt{d} + \sqrt{p}}{\nu} ,$$

where the last inequality follows from the Property 4 of Gaussian random matrix. $\blacksquare$