[Reviews · NeurIPS 2016]

Reviewer 1

Summary

The authors introduce new analytical tools for a class of problems involving matrix estimation from noisy linear measurements. Specifically, they study a regularized least-squares optimization problem that they call the “Generalized Dantzig Selector” (where the regularizer can be an arbitrary norm) and derive upper bounds on the estimation error in terms of certain intuitive geometric parameters, such as the Gaussian width of a certain cone in the matrix space. For the special case when (i) the regularizer is a unitarily-invariant norm (e.g. the Schatten norms) and (ii) the measurement functions as well as the noise are i.i.d. subGaussian, the authors show that the bounds can further be simplified in terms of the size and rank of the underlying matrix. The authors instantiate their bounds for two specific regularizers -- the trace norm and the spectral k-support norm -- and match or improve upon the best previously known existing bounds on the sample complexity of the corresponding optimization method.

Qualitative Assessment

The paper is generally well-written and provides a succinct, self-contained description of the results, techniques, and proofs. Regularized least-squares matrix estimation (with the trace/nuclear norm as the regularizer) is a well-studied problem; however, the authors address the more challenging case of sub-Gaussian observations and when the regularizer cannot be easily decomposed. For such problems, they provide an interesting solution using tools such as generic chaining, and it is possible that such methods can be useful for other kinds of matrix estimation problems as well. My main comment is that the paper could benefit from more exposition and discussion. Particularly, the relation with prior existing methods is a bit unclear from the current manuscript, and potentially can obscure the novelty and impact of the proposed results. To my knowledge, the sample complexity bound for trace-norm regularization (n > \sqrt{r(d+p-r)}) was very well established, so the results of Section 5.1 aren’t too novel. The sample complexity bound for spectral k-support norm-regularization in Section 5.2 seems to be new and interesting, but it is unclear what advantages this particular norm has over other existing norms in terms of estimation quality. Finally, the authors could comment on the applicability of their methods for other kinds of structural constraints, such as general atomic norms.

Confidence in this Review

2-Confident (read it all; understood it all reasonably well)


Reviewer 2

Summary

This paper analyses the performance of the generalized Dantzig selector for the problem of structured matrix recovery. The authors provide deterministic recovery guarantees, and error bounds for sub-Gaussian noise and measurements. The latter bounds are expressed via geometric quantities such as Gaussian widths, for which further bounds in terms of the rank r, and dimensions d and p are provided. The analysis and a bound on matrix recovery using the spectral k-support norm appear to be novel.

Qualitative Assessment

I find the results of the paper interesting and the analysis non-trivial. That being said however, I think the paper is not organized in a reader friendly manner. There are multiple typos, and in Section 4 definitions and lemmas can be moved to the supplement to avoid distracting the reader. I will detail my suggestions below. Re-organization: This suggestion is regarding reorganization of Section 4.1. In my opinion, Lemma 2 and the introduction of the OWL seminorm is completely distracting, since it is not at all apparent what is the significance of these concepts at first glance. Only after reading the proofs I understood why they are introduced. Note that these definitions are not needed for the statements of the Theorems except for the definition of rho. In fact they only appear briefly in the proof of Theorem 6. I suggest these to be moved to the supplement. In addition the proof of Theorem 6 can also be moved to the supplement. I would appreciate to see another example, a data experiment or more discussion in the main paper instead. Finally, the definition of rho, as it is, is also not completely satisfactory. You can define rho as the maximal among all sub-gradients theta^* ratio between the largest and smallest elements . Clarifications: line 100: Lemma 1, Perhaps you want to clarify that f^* is the dual norm to f. line 127: define polar cone line 177: define seminorm In the proof of Theorem 3 line 75 please include a proof why the rank of delta is at most 2r. line 233-234: what is z in the definition of the set D? Perhaps you mean theta^*? lines 307-309: it is not clear from what is said here exactly how are bounds from [16] related to the novel bound. Minor typos: line 22: has been investigated -> have been investigated line 146: supreme -> supremum line 187: will induce -> induces line 195: Base on -> Based on

Confidence in this Review

2-Confident (read it all; understood it all reasonably well)


Reviewer 3

Summary

Authors study the matrix estimation problem via Dantzig selector. They borrow ideas from high dimensional structured parameter estimation literature and apply them to more matrix specific problems. They obtain results in terms of Gaussian width and restricted strong convexity when samples are subgaussian by making use of generic chaining techniques developed by Mendelson.

Qualitative Assessment

This paper borrows standard ideas that were developed during the last 5 years or so to provide guarantees for matrix estimation. One of the main contribution is that authors study multiple matrix related norms and their associated geometry. The other main contribution is that they can give guarantees for subgaussian samples. Overall, the paper looks accurate, but lacks novelty. Both low-rank recovery subject and sub-gaussian sampling are well studied and understood hence the contribution is rather incremental. It might also be a better fit for a signal processing journal rather than NIPS. Regarding subgaussian measurements, in page 2, the authors state: "Related advances can be found in [...], but they all rely on the Gaussian measurements to which classical concentration results [15] are directly applicable." An important issue is that handling subgaussian measurements is NOT a contribution of this paper. It is merely an application of what is already out there due to Mendelson (and a few more researchers). While this is stated in Theorem 4, introduction makes it into a major contribution. In fact, one can obtain identical results for subexponential or even subsampled Fourier measurements (Fast JL Trasform) all in terms of Gaussian width. Authors also do not refer to a *very* related paper ``A simplified approach to recovery conditions for low rank matrices''. This paper connects matrix and vector estimation problems similar to what authors are trying to do. They should also consider citing related low-rank estimation papers such as ``Null space conditions and thresholds for rank minimization'' and chaining papers such as ``Toward a unified theory of sparse dimensionality reduction in euclidean space''.

Confidence in this Review

3-Expert (read the paper in detail, know the area, quite certain of my opinion)


Reviewer 4

Summary

The paper considers the unified analysis on the structured matrix estimation via the generalized Dantzig selector. The paper focuses on the L2 (or Frobenius) estimation upper bound for the sub-Gaussian designs.

Qualitative Assessment

My main concern is on its novelty. My impression is that even the unitary invariant norms (especially two examples, trace norm and spectral k-support norm) can be understood in the framework of [22] with decomposable norms. Is there any other example that is covered by the proposed framework but not by [22]? Admittedly, deriving corollaries for matrix recovery estimators with spectral k-sup etc. are also not trivial, but if they are the special cases of the existing framework, the contributions of this paper might be degraded. Minor comments: - Eq (9) ends with comma. - I think the proofs in the paper looks a bit cluttering. It might be better to move them to the appendix as other proofs, and add more discussion/comparisons with the saved space.

Confidence in this Review

2-Confident (read it all; understood it all reasonably well)


Reviewer 5

Summary

This paper studies the problem of structured matrix recovery via Generalized Dantzig Selector. It also provided some examples of the general bound by applying it to unitarily invariant norms, including the trace norm and spectral k-support norm.

Qualitative Assessment

Comments: 1. The presentation of this paper needs to be improved. Section 2 should be put after Section 3, since Section 3 contains many background knowledge. In addition, the whole Section 3 should be simplified, and most of them can be moved into the appendix, since these are cited from existing literature, and are not the contribution of this paper. 2. There is no intuition behind most proofs in Section 4. Most of the proofs are very technical and lack of insights. The authors should first give a high-level proof roadmap, and then prove each step. For those proofs without any insights (e.g., Proofs of Theorems 4,6,7), the authors should either add more comments/remarks, or just move them into the appendix. 3. My major concern is the novelty of this paper. This paper combines a lot of existing known results, from restricted strong convexity based proof technique for Dantzig type estimator (Theorems 1 and 4), to bounding Gaussian width of different geometric structures using generic chaining (Theorem 6 and 7). However, the resulting bounds for matrix recovery by minimizing different norms (e.g., nuclear norm and spectral k-support norm) are not sharper than existing results at all. What can we gain by using Matrix Dantzig selector? Or using Gaussian-width based argument? The only advantage I can imagine is the geometric interpretation behind Gaussian width, but this does not lead to any advancement in algorithmic design. 4. The authors claimed that they derive general bounds on those geometric measures for structures characterized by unitarily invariant norms, a large family covering most matrix norms of practical interest. 
However, in Section 5, they only show two examples (nuclear norm and spectral k-support norm), which are well studied and not very interesting. Any other examples? == post-rebuttal update== I have read the authors' rebuttal.

Confidence in this Review

2-Confident (read it all; understood it all reasonably well)


Reviewer 6

Summary

The paper under review consider recovery of general structured measurements with the Danzig selector from linear measurements. Specifically, the paper assumes measurements y_i = < \theta,X_i > + w_i, where X_i are measurement matrices and w_i is a scalar corresponding to noise. The paper considers the estimator arg min_\theta R(\theta) subject to R*(\sum_i ( < \theta,X_i > - y_i ) X_i ) \leq \lambda_n, where R is an arbitrary matrix norm and R* is its dual. The paper assumes sub-Gaussian measurement matrices and sub-Gaussian noise, which is a very general setup. The main result of the paper appears to be what is stated in the text in the first paragraph of Section 5: If the number of measurements is larger than the squared Gaussian width of the tangent cone with respect to the norm R, then the recovery error is bounded in a certain sense. This is similar to [Cor. 3.3, 10], which pertains to Gaussian measurements and bounded noise.

Qualitative Assessment

1. In essence the paper provides generalizations of known results: It shows that matrices with structure (e.g., low rank) can be recovered from sub-Gaussian measurements. What appears to be new is the very general problem setup. Specifically the paper considers general norms R as well as the measurement matrices and the noise are assumed to be sub-Gaussian instead of Gaussian, for which corresponding results are already available. However, several works have shown already that Gaussian measurement matrices can be substituted by sub-Gaussian measurement matrices; this is essentially a consequence of theory developed in [10] and Gordan's lemma holding for more general than Gaussian matrices. See for example the work by Dirksen ``Dimensionality Reduction with Subgaussian Matrices" for a concrete example of this line of work. While I understand that it is not easy to prove such results, the paper would need to explain if and why the results are novel with respect to the literature mentioned above, and why and if they provide a substantial improvement with respect to the state of the art. It might be for example that there are interesting choices of the norm R that are relevant and not covered by results in the literature yet. 2. After having read the introduction several times, it is not clear what the main contribution and the specific problem considered of the paper is. Specifically, the paper does not elaborate what it means with ``generally structured matrices''. As a consequence, it is not clear whether the problem of recovering ``generally structured matrices'' includes interesting examples beyond the usual low rank models covered by the literature already. 3. In general the writing could be improved significantly; the paper contains many typos (e.g., only on the bottom of page 5: Lemma 2 ends without colon, Base-> Based; supplementary -> supplementary material; etc), and more importantly, the statements have to be combined by the reader himself to obtain the most interesting results of the paper. Specifically, the current structure is the following: Section 2 starts with a deterministic recovery condition (Theorem 1) that expresses the estimation error in Frobenius norm in terms of the geometry of the tangent cone at \theta*. Section 4 bounds quantities relevant for the deterministic recovery condition in Theorem 1 for sub-Gaussian measurement matrices and sub-Gaussian noise. Finally, in Section 5, the paper states in words what the results in Section 2 and 4 combined yield, and some implications. This is one of the most interesting conclusions of the paper, and appears to be the main result of the paper (see above). It would be great if the paper would be reorganized so that a main result, e.g., what is stated in the first paragraph of Section 5, would be stated and discussed in detail early in the paper. Moreover, additional explanations on how this result improves the state of the art, and pointing to special case already treated by the literature (e.g. [Cor. 3.3, 10]) would be helpful. As it is now, Section 4 contains several theorems that bound and relate quantities relevant for the deterministic recovery condition. E.g., Theorem 4 lower bounds the condition in terms of the Gaussian width of the tangent cone (note that this result uses generic chaining and is stated for subgaussian matrices; however for Gaussian matrices this is well know, see [10]) Only combined with Theorem 1 those statements yield a recovery result; if this is what the paper sets out to do, then it would be significantly more clear if all those Theorems 3-7 were stated as lemmas, and only one central Theorem were stated and discussed. Minor comments: 1. Sec. 2: ``The recovery bound assumes not knowledge of R'': It is not clear what is meant with this statement. The recovery bound depends on the norm R, and the convex program assumes knowledge of R. 2. Sec. 4 ``X_i's are copies of a random vector'': Those are matrices.

Confidence in this Review

2-Confident (read it all; understood it all reasonably well)